# Biotechnological Key Genes of the *Rhodococcus erythropolis* MGMM8 Genome: Genes for Bioremediation, Antibiotics, Plant Protection, and Growth Stimulation

**DOI:** 10.3390/microorganisms12010088

**Published:** 2023-12-31

**Authors:** Daniel Mawuena Afordoanyi, Yaw Abayie Akosah, Lidiya Shnakhova, Keremli Saparmyradov, Roderic Gilles Claret Diabankana, Shamil Validov

**Affiliations:** 1Laboratory of Molecular Genetics and Microbiology Methods, Kazan Scientific Center of Russian Academy of Sciences, 420111 Kazan, Russiar.diabankana@knc.ru (R.G.C.D.);; 2Tatar Scientific Research Institute of Agricultural Chemistry and Soil Science, FRC Kazan Scientific Center, Russian Academy of Sciences, 420111 Kazan, Russia; 3Department of Molecular Pathobiology, College of Dentistry, New York University, New York, NY 10010, USA; 4Dermatology Department, I.M. Sechenov First Moscow State Medical University (Sechenov University), 119991 Moscow, Russia

**Keywords:** antiSMASH, *Rhodococcus erythropolis*, full genome, secondary metabolites, resistant genes, pan-genome

## Abstract

Anthropogenic pollution, including residues from the green revolution initially aimed at addressing food security and healthcare, has paradoxically exacerbated environmental challenges. The transition towards comprehensive green biotechnology and bioremediation, achieved with lower financial investment, hinges on microbial biotechnology, with the *Rhodococcus* genus emerging as a promising contender. The significance of fully annotating genome sequences lies in comprehending strain constituents, devising experimental protocols, and strategically deploying these strains to address pertinent issues using pivotal genes. This study revolves around *Rhodococcus erythropolis* MGMM8, an associate of winter wheat plants in the rhizosphere. Through the annotation of its chromosomal genome and subsequent comparison with other strains, its potential applications were explored. Using the antiSMASH server, 19 gene clusters were predicted, encompassing genes responsible for antibiotics and siderophores. Antibiotic resistance evaluation via the Comprehensive Antibiotic Resistance Database (CARD) identified five genes (*vanW*, *vanY*, *RbpA*, *iri*, and *folC*) that were parallel to strain CCM2595. Leveraging the NCBI Prokaryotic Genome Annotation Pipeline (PGAP) for biodegradation, heavy metal resistance, and remediation genes, the presence of chlorimuron-ethyl, formaldehyde, benzene-desulfurization degradation genes, and heavy metal-related genes (*ACR3*, *arsC*, *corA*, *DsbA*, *modA*, and *recG*) in MGMM8 was confirmed. Furthermore, quorum-quenching signal genes, critical for curbing biofilm formation and virulence elicited by quorum-sensing in pathogens, were also discerned within MGMM8’s genome. In light of these predictions, the novel isolate MGMM8 warrants phenotypic assessment to gauge its potential in biocontrol and bioremediation. This evaluation extends to isolating active compounds for potential antimicrobial activities against pathogenic microorganisms. The comprehensive genome annotation process has facilitated the genetic characterization of MGMM8 and has solidified its potential as a biotechnological strain to address global anthropogenic predicaments.

## 1. Introduction

The accumulation of contaminants from industrial, pharmaceutical, and agricultural malpractices has harmed the environment. Most of the chemicals that are meant to solve the problems subsequently lead to complications that demand intensive alternative solutions. Bioremediation, alternative microbial antibiotics, and biocontrol agents with plant growth-promoting abilities are the biotechnological approaches that have been adopted in eradicating industrial contaminants, antibiotic-resistant bacteria, and agricultural chemicals [1]. One of the biotechnological bacterial genera that have been deployed for this purpose is the actinomycetes genus *Rhodoccoccus,* based on its active secondary metabolites relevant in the above-mentioned sectors [2,3].

Against phytopathogenic fungi and bacteria, strains of *Rhodococcus* have the mechanism of antibiosis, and aflatoxin degradation to prevent the formation of biofilm through quorum-quenching [3,4,5]. Taking into consideration the key properties for its use as a bioremediation microbe, *Rhodococcus* spp. has been documented to adapt to hydrocarbon assimilation, ecological plasticity, biosafety in relation to humans, and also its effectiveness in situ, on-site, and during bioreactor treatment [6]. *Rhodococcus erythropolis* is an *Actinobacterium* with the versatility to grow in adverse abiotic and biotic stress conditions, including the wide range of complex compounds that it degrades [7,8,9]. Highly toxic and flammable, formaldehyde from a phenolic and melamine resin-manufacturing company was degraded and removed by *R. erythropolis* UPV-1 [10]. The residue of the sulfonylurea herbicide chlorimuron-ethyl is another extreme contaminant that was detoxified by the strain *R. erythropolis* D310-1 thanks to the key gene *carE* [11]. In the pharmaceutical sector, the non-ribosomal peptides that are synthesized have high antimicrobial activity, and the most fascinating data are reported by Hu et al. [12] regarding the exopolysaccharide produced by *R. erythropolis* HX-2 that inhibited the growth of cancer cells.

In general, the data regarding their biotechnological potential are extensive, and the variations in strains are based on the environments in which they are isolated, resulting in distinct strain properties. Bioinformatics tools have been instrumental in annotating and predicting the unique genes involved in these phenotypic characteristics. This approach helps to reduce trial-and-error processes and “blind” experimental setups that rely on the phenotypic properties of other strains. Most *R. erythropolis* strains’ genomes typically include plasmids responsible for enhancing the strains’ adaptability to severe conditions, as reported in *Agrobacterium* [13]. The annotation of the chromosomal genomes of different strains predicted genes related to activities such as quorum-quenching (the lactonase-encoding gene *qsdA*), degradation of catechol, 3-nitrotoluene, phthalate, and benzoate (through the operons *pheRA2A1* and *catRABC*), breakdown of sterol molecules (involving oxygenase KshA and reductase KshB), and desulfurization of fuel molecules (including *dszA*, *dszB*, *dszC*, and flavin reductase *dszD*) [14]. Interestingly, the authors explained that most of the genes responsible for cleaving carbon–sulfur bonds in fuel molecules (*dszA*, *dszB*, and *dszC*) are usually found on the plasmids of *R. erythropolis*, whereas *dszD* is associated with the chromosomal genome [14].

*Rhodococcus erythropolis* strain MGMM8 is a novel strain isolated from the rhizosphere of a winter wheat plant that can be included in the already-known biotechnological strains. For the further determination of its application, we isolated and sequenced the chromosomal genome of the strain to determine the genes responsible for bioremediation, antibiotics, and its use in different types of biotechnology such as medical, industrial, and environmental biotechnology.

## 2. Materials and Methods

### 2.1. Isolation of Strain from Rhizosphere

For the isolation of *R. erythropolis*, 100 mg of the rhizosphere of the winter wheat plant was weighed into a 1.5 mL Eppendorf tube containing 900 µL phosphate-buffered saline (PBS) (140 mM NaCl, 5 mM KH_2_PO_4_, 1 mM NaHCO_3_, pH 7.4) and vortexed for 30 s. Serial dilution up to 10^−6^ was performed and 100 µL was then pipetted onto a Petri dish with Gauze’s synthetic medium no. 1 (DSMZ, Braunschweig, Germany) and Gauze’s synthetic medium no. 2 (DSMZ, Braunschweig, Germany). The culture was gently spread with a glass spreader and placed in a thermostat to grow at 30 °C for 72 h. Single colonies were then picked and replated for DNA isolation 16S rRNA analysis and subsequent full-genome sequencing. To determine the morphological characteristics of the strain, overnight culture was streaked on the above media including Luria–Bertani (LB) agar medium (g/L: tryptone, 10 g; yeast extract, 5 g; NaCl, 10 g; agar, 15 g) with the help of an inoculating loop.

### 2.2. DNA Isolation and Identification

The isolation of total DNA from bacterial strain was performed using the TRIzol kit (Invitrogen, Carlsbad, CA, USA) following the manufacturer’s protocol. The amplification of the conservative gene 16S rRNA was carried out using the PCR method in accordance with the recommendation by Weisburg et al. [15]. The amplificated fragment was excised and purified from 1.5% gel agarose using a Cleanup Mini kit (Evrogen, Moscow, Russia). The cleaned DNA was quantified and sent to Evrogen for sequencing. The obtained 16S rRNA sequence chromatogram was searched using the National Center for Biotechnology Information (NCBI) Basic Local Alignment Search Tool (BLAST) for identification.

### 2.3. Genome Annotation of R. erythropolis MGMM8

For species-affiliation confirmation and a full-genome annotation of *R. erythropolis* MGMM8, total DNA was isolated from an overnight culture as described in Section 2.2. The DNA was then sequenced on Illumina HiSeq 2500 with 2 × 125 bp paired-end reads at CeGaT Biotechnology Co., Ltd. (Tübingen, Germany). To assemble the genome from contigs, the quality was checked using FastQC (v. 0.11.2) [16]. The trimming of adapters and low-quality reads was performed using Trimmomatic v. 0.36. [17]. To assemble the genome de novo, Spades v. 3.12 and Unicycler v. 0.5.0 were used [18,19]. The preliminary assembled genome was then blasted for closely related genomes using FASTANI [20], ANIb (average nucleotide identity based on BLAST), and JspeciesWS (https://jspecies.ribohost.com/jspeciesws/#home, accessed 13 April 2023). The rearrangement of contigs was then conducted again using SCAR-web (webserver for contigs rearrangements) [21] with a subsequent filling and closing of gaps between contigs using GAPPadder v. 1.10 [22] and GapBlaster v. 1.1.1 [23].

For the prediction of secondary metabolite biosynthetic gene clusters, the bioinformatic server antiSMASH v. 7.0 [24] was used and compared with the complete genomes of four *R. erythropolis* strains (JCM2895, X5, D310-1, CCM2595) from the NCBI database, with accession numbers AP018733.1, CP044284.1, NZ_CP032403.1, and NC_022115.1, respectively. Likewise, their antimicrobial-resistant genes were predicted using the Comprehensive Antibiotic Resistance Database (CARD) [25]. Key genes for the degradation of contaminants were predicted using the automatic system NCBI Prokaryotic Genome Annotation Pipeline (PGAP) (http://www.ncbi.nlm.nih.gov/genome/annotation_prok, accessed on 17 August 2023). OrthoVenn3 was used to compare the orthologous protein groups of selected species of *R. erythropolis* strains [26].

### 2.4. Screening the Potential of R. erythropolis MGMM8 to Produce Biosurfactant

#### 2.4.1. Cell-Free Preparation

The cell-free preparation of MGMM8 was undertaken from its overnight cell culture grown on fermentation medium [10 g/L unrefined sunflower oil; 10 g/L solution #1 (1% peptone, 4.75 g/L K_2_HPO_4_, 1 g/L NH_4_Cl, 6 g/L MgSO_4_ × 7H_2_O) and 1 mL of solution #2 (100 mg FeSO_4_ × 7H_2_O, 100 mg MnCl_2_ × 4H_2_O, 100 mg ZnSO_4_ × H_2_O, and 100 mg of CaCl_2_ × H_2_O) pH 7.5]. For this purpose, the culture was centrifuged at 5000 rpm for 5 min. The obtained supernatant was passed through a 0.45 μm Pore Size membrane filter (Merck, Darmstadt, Germany).

#### 2.4.2. Drop Collapse Assay

The ability of MGMM8 to produce a biosurfactant was evaluated based on the ability of its cell-free supernatant to decrease the surface tension between water and hydrophobic surfaces. For this purpose, 50 µL of extracted cell-free supernatant was dropped onto the Parafilm M^®^ (Heathrow Scientific, Vernon Hills, IL, USA) surface. The process of droplet flattening and spreading on the parafilm was observed for up to 4 min. Subsequently, methylene blue was added to the stain, which did not affect the droplet’s shape. The mixture was allowed to dry, and the diameter of the dried droplet was measured.

#### 2.4.3. Oil Displacement Test

The oil displacement test was conducted following the method described by Ohno et al. [27] and Rodrigues et al. [28]. For this purpose, 35 mL of tap water was poured into a 90 mm Petri dish. Subsequently, 500 μL of crude petroleum oil was carefully added to the water surface, followed by the addition of 100 μL of a cell-free culture supernatant of MGMM6 onto the oil surface. After incubation, the diameter of the clear zone formed was measured. Each experiment was repeated twice for statistical analysis, and each group was maintained for three repetitions. Sodium dodecyl sulphate (SDS) solution (0.5%, *w*/*v*) was used as a positive control. The experiment was repeated twice for statistical analysis, and each sample was analyzed in triplicate.

#### 2.4.4. Bio-Emulsification Index

The emulsification rate of MGMM8 was measured according to Cooper and Goldenberg [29]. For this purpose, the cell-free supernatant of MGMM8 was mixed in a ratio (1:1) with petroleum oil. The mixture was homogenized for 3 min at 28 °C. After homogenization, the obtained mixture was incubated for up to 24 h at room temperature. The emulsification index (E24%) was measured using the following formula:E24% = (height of the emulsifying layer/the total height solution) × 100(1)

## 3. Results

### 3.1. Morphological Characteristics and Genomic Features of R. erythropolis MGMM8

The morphology of the colonies exhibited distinct features. On the LB medium (Figure 1A), they appeared as circular formations, measuring approximately 2 mm in diameter, displaying a glossy beige hue, well-defined solid edges, and a slightly mucoid consistency. Conversely, when cultivated on Gauze’s synthetic medium no. 1 (Figure 1B), the colonies exhibited smaller dimensions, with an average diameter of approximately 1 mm. They maintained solid, round edges and retained a mucoid texture to some degree. On Gauze’s synthetic medium no. 2, these colonies demonstrated pointwise growth, characterized by compact, solid, and round edges, with a predominantly white coloration (Figure 1C).

The full genome was deposited into the NCBI database under the accession number CP124545.2 with a circular chromosome of 6,376,501-bp comprising 6132 total genes and 6062 CDS (with proteins). The genome also contains 60 genes (RNA), 52 tRNAs, 3 ncRNA, 70 pseudogenes (total), and 62% G + C content.

### 3.2. Comparison of Secondary Metabolite Gene Clusters in Chromosomal Genomes

The secondary metabolite gene cluster prediction performed on the antiSMASH bioinformatic server showed that *R. erythropolis* MGMM8 (Table 1) had 19 gene clusters with 3 unidentified gene clusters of 2 different Type I polyketide synthase (T1PKS) and 1 lanthipeptide-class-iii type gene cluster. In comparison with the chromosomal genomes of the four published strains, the antiSMASH server predicted a 100% similarity to non-alpha poly-amino acids (NAPAA) ε-Poly-l-lysine, non-ribosomal peptide synthetase (NRPS) corynecin III/corynecin I/corynecin II, and NRP-metallophore NRPS heterobactin A/heterobactin S gene clusters in all five strains. The gene cluster for the anabolism of a post-translationally modified peptide product (RiPP-like) for branched-chain fatty acids and the osmolyte estoine showed a 75% homology in all the compared genomes. Other similarities among the compared strains include the prediction of a linear azol(in)e-containing peptide (LAP) type for the diisonitrile antibiotic SF2768, a T1PKS encoding fulvuthiacene A/fulvuthiacene B, NRP-metallophore erythrochelin, and NRPS terpene for tetracycline SF2575 cluster gene with percentages of 11%, 8%, 57%, and 6%, respectively. The biosynthetic gene cluster for the carboxylic polyether ionophore monensin, with a similarity of 5%, was predicted in all compared strains except D310-1. Also, the gene cluster for the biosynthesis of carotenoid (27% similarity) was found in all strains except for CCM2595, which had a 37% similarity. 

In terms of unique gene clusters, an NRPS-like sulfur-containing antibiotic, thiolutin, was predicted in the chromosomal genomes of MGMM8 and JCM2895. The lowest percentage similarity of the gene clusters was predicted for the redox-cofactor type (tetronasin). However, NRPS-type rifamorpholine A/rifamorpholine B/rifamorpholine C/rifamorpholine D/rifamorpholine E were not predicted in MGMM8. One interesting secondary metabolite cluster gene found in MGMM8, D310-1, and CCM2595 was the PKS-like, amglyccycl, responsible for the biosynthesis of the acarbose pseudo-tetrasaccharide, which is an alpha-glucosidase-like inhibitor for the treatment of type 2 diabetes with a gene similarity of 7%. The main annotation results can be seen in the Appendix A.

### 3.3. Comparison of Antimicrobial Genes Clusters in Chromosomal Genomes

The results for the prediction of antimicrobial genes performed on the CARD server showed the presence of a minimum of four gene families in each chromosomal genome annotated on the server (Table 2). Interestingly, the same five resistant genes were predicted in both MGMM8 and CCM2595 from VanW, glycopeptide resistance gene cluster, VanY, glycopeptide resistance gene cluster, rifampin monooxygenase, RbpA bacterial RNA polymerase-binding protein, and aminosalicylate resistant dihydrofolate synthase AMR gene families. These gene families provide resistance to the drug classes glycopeptide antibiotic, rifamycin antibiotic, and salicylic acid antibiotic using the mechanisms of antibiotic target alteration and antibiotic inactivation (Appendix A). The chromosomal genomes of the strains JCM2895, X5, and D310-1 contained four identical resistant genes from all predicted genes except VanY, glycopeptide resistance gene cluster, and the rifampin monooxygenase resistant gene, *iri* (Table 2).

### 3.4. Annotation of Key Genes in R. erythropolis MGMM8 Involved in the Degradation of Contaminants

The deposition and annotation of the genome into the NCBI database enabled the identification of some key genes responsible for the biodegradation of xenobiotics and contaminants, as well as quorum-quenching and heavy metal-resistant genes in *R. erythropolis* MGMM8. Five carboxylase, twenty-seven amidohydrolase, and one FMN-dependent monooxygenase genes responsible for the chlorimuron-ethyl degradation were predicted in the genome. For the degradation of formaldehyde, two genes encoding glutathione-independent formaldehyde hydrogenase, one gene encoding mycothiol-dependent formaldehyde hydrogenase, and methanol hydrogenase (mdo), which potentially acts as formaldehyde dismutase [26], were identified. Concerning genes responsible for quorum-quenching, *R. erythropolis* MGMM8 harbors the genes for 3-oxoadipate enol-lactonase and FadD3 family acyl-CoA ligase. Another interesting result was the presence of two flavin reductase (dszD) genes and six flavin reductase family protein genes, which are essential for the biodesulfurization of fuel molecules. Out of the 2 flavin reductase predicted, the first gene with 75 amino acids (aa) located at the 5,140,661–5,140,888 bp region appeared to be incomplete. Notwithstanding, in silico Protein Blast on the NCBI BLAST server showed a 100% Query Cover and 100% for MULTISPECIES: flavin reductase [*Rhodococcus*], with the accession number WP_223257974.1 (accessed on 31 August 2023). The second gene with 190 aa (5,432,346 bp–5,432,918 bp) also showed 100% identity to the flavin reductase [*Rhodococcus erythropolis*], with the accession WP_046379200.1.

The genes ACR3 and arcS, which provide resistance to arsenic (As) and antimony (Sb), were also predicted in the genome of MGMM8. The genome also harbors the gene corA, which encodes a transporter protein for the metals magnesium (Mg), cobalt (Co), nickel (Ni), and manganese (Mn). For resistance to the metals cadmium (Cd), zinc (Zn), and mercury (Hg), the gene encoding the DsbA family protein was present in the chromosomal genome of MGMM8. The gene for molybdate ion transporter protein modA, which also transports tungsten (W), was also found in the genome. Finally, the ATP-dependent DNA helicase gene recG present in the genome is said to confer resistance to the metals chromium (Cr), tellurium (Te), and selenium (Se).

A more focused orthology analysis of *R. erothopholis* MGMM8 compared with related strains using OrthoVenn 3 (Figure 2) was conducted. These analyzed genome strains shared 5184 clusters of orthologous proteins, indicating shared protein function (Figure 2A). Only 15 clusters (including glycerophosphodiester transport (3), monoterpenoid metabolic process (2), metal ion binding (2), regulation of transcription (4), and transmembrane transport (4) clusters) were unique to MGMM8 (Figure 1C). The genome of D310 carried nine clusters of orthologous proteins, including catabolic process (2), metal ion binding (2); fatty acid biosynthetic process (2); and extracellular region (3), which are involved in biological processes such as organic acid metabolic process, lipid metabolic process, and cellular lipid metabolic process. The genome of X5 encodes two unique clusters of orthologous proteins involved in the compound catabolic process. Seven (7) clusters of orthologous proteins (including DNA restriction-modification system (2), DNA restriction-modification system (2), DNA restriction-modification system (2) and transposition, DNA-mediated (1)) involved in biological processes such as DNA restriction-modification system and transposition and DNA mediation were unique to JCM2895. The genome CCM2595 encodes two unique hypothetical homologous protein groups. The analyzed genomics of X5, D310, JCM2895, MGMM8, and CCM2595 carried 246, 409, 157, 290, and 110 singleton proteins (proteins that never cluster with one another), respectively (Figure 2B).

### 3.5. Biosurfactant Production of R. erythropolis MGMM8

The obtained results (Figure 3 and Figure 4) showed that *R. erythropolis* MGMM8 can produce biosurfactants. It can be observed (Figure 3) that the clear zone formed by the free-cell supernatant of MGMM8 on the surface of crude oil was larger, measuring approximately 6.73 ± 0.84 cm in diameter (Figure 3C), compared to the clear zone formed by 0.5% SDS, which developed a diameter of 6.31 ± 0.53 mm (Figure 3B). No statistical difference between MGMM8 free-cell and supernatant and 0.5% SDS solution was noted in terms of the formed clear zone.

In addition, MGMM8 exhibited effective bioemulsification activity when tested with unrefined sunflower oil (Figure 4). The formation of two separate layers that differed in their heights was observed (Figure 4A-(1)), compared with the control (Figure 4A-(2)). The emulsification index was measured as 45.18 ± 0.23%. An identical result was obtained in terms of drop collapse (Figure 4B). The drop diameter of cell-free surfactant produced by MGMM8 (Figure 4B-(3)) was greater than that produced by SDS solution (1% *w*/*v*) (Figure 4B-(4)) and control (Figure 4B-(5)). The diameter of the cell-free supernatant of MGMM8 diameter was measured as 0.97 ± 0.01 cm and 0.82 ± 0.04, respectively. These obtained results indicated that *R. erythropolis* MGMM8 produce biosurfactant.

## 4. Discussion

Full-genome annotation helps us to predict the genetic characteristics of a strain for a complete overview of its biotechnological applications. Based on their multiple catabolic genes and the high genetic redundancy of biosynthetic and degradation pathways, *Rhodococcus* spp. have been used in plant protection, anticancer inhibition, and bioremediation [2,3,12,30]. In this study, we were able to identify gene clusters and key genes *of R. erythropolis* MGMM8 for its experimental phenotypic characteristics and applications in pharmaceutical, agricultural, and ecological sectors. This includes antimicrobial secondary metabolite annotation, genes for antibiotic resistance, quorum-quenching genes, biodegradation genes for contaminants such as chlorimuron-ethyl, formaldehyde, and genes for resistance to heavy metals and their remediation. Primarily, the antiSMASH server’s percentage similarity prediction focuses solely on the KnownClusterBlast platform within its supplementary attributes. This platform pertains to extensively annotated gene clusters sourced from the Minimum Information about a Biosynthetic Gene Cluster (MIBiG) dataset (Appendix A). Conversely, the ClusterBlast feature exhibits complete homology (100% identity) to clusters originating from distinct strains of identical species (Appendix A). This underscores the potential for a comprehensive annotation of a significant proportion of these clusters to ascertain their precise identity percentage.

Taking into consideration the gene clusters with 100% identity, the excellent antibiotic ε-poly-l-lysine is a non-toxic, edible, and biodegradable homopolypeptide from microorganisms of the genera *Streptomyces*, *Epichloë*, *Streptoverticillum*, and *Kitasatospora* [31]. The main microorganism that is known to produce this compound *is Streptomyces albulus*, with a recent manuscript describing a high-yield production from the strain WG-608, a mutant generated from M-Z18 [32]. Given the existing genetic alignment within the chosen genome, it becomes imperative to isolate and assess its inherent potential. Furthermore, notable instances encompass the chloramphenicol-like antibiotics: corynecin III, corynecin I, and corynecin II, initially derived from the cultured medium of *Corynebacterium hydrocarboclastus* [33,34]. This natural product was isolated from the supernatant of *Rhodococcus* sp. H-CA8f and analyzed through chemical dereplication using a liquid chromatography-high resolution MS [35]. This was the first report on its isolation from *Rhodococcus* spp., and it will be of great relevance to isolate and identify this antibiotic from MGMM8. Heterobactin, a catecholate-hydroximate mixed-type siderophore with a 100% similarity prediction in all the genomes used in this study has been reported to be associated with *R. erythropolis* [36]. Through their metal-chelating properties, heterobactins play a pivotal role in the collaborative biocontrol efficacy of *R. erythropolis* against phytopathogens. In addition, these molecules have demonstrated an affinity for binding to arsenic [37,38]. The aforementioned details underscore the prospective utility of MGMM8 in diverse domains such as medicine, plant protection, and bioremediation. However, for effective deployment in these applications, it is necessary to conduct phenotypic characterization.

The biosynthetic gene clusters (BGCs) displaying 75% similarity encompass various functionalities, including the synthesis of compounds responsible for membrane fluidity maintenance, branched-chain fatty acids, and the osmolyte ectoine (1,4,5,6-tetrahydro-2-methyl-4-pyrimidine carboxylic acid), which is vital for halophilic microorganism adaptation [39,40]. Notably, ectoine serves as a structural component facilitating survival under unfavorable conditions, with its presence confirmed in the genome of *R. erythropolis* MGMM8. Considering its significance, conducting a halophilic test to assess this property assumes relevance for genetic manipulation in other microbes, as indicated by Ning et al. [41]. Other BGCs, displaying similarity levels below 60%, encompass diverse functionalities. Among them are tetrapeptide hydroxamate-type siderophores (erythrochelin and coelichelin), which exhibit inhibitory effects against *Bacillus*, *Micrococcus* spp., and *Pseudomonas aeruginosa* under iron-deficient conditions [42,43]. Moreover, carotenoid production, noted to significantly increase in *R. erythropolis* JCM3201 exposed to white light, continues to be expressed [44]. This phenomenon has been elucidated by Dat et al. [45], who attribute it to certain BGCs possibly responsible for producing analogous or identical compounds. This analysis was conducted using the Biosynthetic Genes Similarity Clustering and Prospecting Engine (BIG-SCAPE), which groups homologous BGCs into gene cluster families (GCFs). Plausible biosynthetic activities may underlie the remaining predicted BGCs within the MGMM8 genome. Regarding the CARD results, two prominent resistance genes are evident in *R. erythropolis* strains: the *VanW* and *RbpA* genes. The presence of five resistance genes in the genome underscores its resilience against xenobiotics or antibiotics within microbial populations of application. Based on the prevalence of *RbpA* in 52 *Rhodococcus equi* genomes [46], we posit this gene as an original evolutionary acquisition after the divergence of *Rhodococcus* spp.

The genome of MGMM8 exhibits a rich complement of genes responsible for both organic and inorganic compound degradation, indicating its substantial bioremediation potential. For the breakdown of chlorimuron-ethyl herbicide residues, the genome contains 27 amidohydrolase (*amiH*) genes, 5 carboxylesterase (*carE*) genes, and FMN-dependent monooxygenase (*Fmo*), the latter of which has been previously characterized in *R. erythropolis* D310-1 [11]. Notably, Lessmeier et al. [47] demonstrated that NAD-linked mycothiol-dependent formaldehyde dehydrogenase not only participates in formaldehyde degradation but also confers tolerance to external stress in *Corynebacterium glutamicum*. Regarding *Rhodococcus* spp., evidence exists for *R. erythropolis’s* proficiency in benzo[α]pyrene utilization and formaldehyde degradation [48]. The presence of three distinct genes responsible for formaldehyde degradation within the genome underscores MGMM8′s capability to remediate contaminants.

In the context of quorum-quenching to inhibit pathogens, 3-oxoadipate enol-lactonase has been identified as the agent responsible for degrading N-acyl homoserine lactones, which are molecules pivotal for biofilm formation and virulence [5]. Furthermore, a novel *fadT* gene encoding acyl-CoA dehydrogenase has been linked to the degradation of diffusible signaling factor (DSF) in *Xanthomonas campestris* pv. *Campestris* [49]. The genome of MGMM8 encompasses numerous genes encoding acyl-CoA dehydrogenase enzymes, suggesting the potential for quorum-quenching activity by the strain, which could be experimentally validated.

Regarding flavin reductase (*dszD*) genes, which are implicated in sulfur-specific reductase activities that cleave carbon–sulfur bonds as described by Kwasiborski et al. [14], the prediction of two genes and six family proteins underscores their substantial involvement in the biodesulfurization of petroleum oil. In contrast to the findings where only one flavin reductase gene was projected in the chromosomal genome of *R. erythropolis* XP (7,229,582 bp) [50], MGMM8 possesses two genes within its chromosomal genome. This variation could be attributed to genomic rearrangements between plasmids and the chromosomal genome.

Another interesting datapoint is the presence of six distinct genes (*ACR3*, *arsC*, *corA*, *DsbA*, *modA*, and *recG*) in the genome of *R. erythropolis* MGMM8, which are associated with the resistance and remediation of heavy metals, accentuating its applicability in the bioremediation of contaminants. As underscored in the review by Nazari et al. [26] on the capabilities of the *Rhodococcus* genus in remediating PAHs (polycyclic aromatic hydrocarbons), phenolic compounds, emerging contaminants, heavy metals, and dyes, the annotation of *R. erythropolis* MGMM8 unveils its potential for such purposes. The orthologous protein clusters analysis reveals the presence of fifteen genes unique to MGMM8 in relation to the four compared chromosomal genomes. Interestingly, there is the presence of the three gene-encoding proteins for glycerophosphodiester transport, which is involved in the supply of inositol, choline, phosphate and glycerol for their metabolism in different biochemical pathways [51].

The most important metabolites for the biotechnological remediation of hazardous anthropogenic pollutions (crude oil chemical base products) are biosurfactants and bio emulsifiers produced by microbes [52,53,54]. The ability of *Rhodococcus* strains to produce biosurfactants has been well-documented [55,56,57,58,59]. To confirm whether MGMM8 has the potential to phenotypically produce amphiphilic molecules, an oil spreading test, emulsification, and drop collapse assay were performed using an unrefined sunflower oil as a hydrophobic substrate. The results revealed a high production of glycolipid biosurfactants, with an E24 emulsion index of approximately 45.18 ± 0.23%. Similar results were previously reported by Sadouk et al. [60] and Pirog et al. [61], whereby the emulsion E25 produced by *R. erythropolis* strains tested reached up to 60% when 2–3% residual sunflower frying oil was used as a substrate. The oil displacement assay is considered a sensitive method for detecting small amounts of produced biosurfactants due to their ability to form a halo zone over the oil surface [62,63,64]. In the current study, the supernatant isolated from MGMM8 also induced the formation of a halo zone on crude petroleum oil. This attests to the ability of MGMM8 to be a candidate for bioremediation. However, further studies are needed to accurately assess their effectiveness in field applications.

## 5. Conclusions

The conventional scientific approach to selecting a biotechnologically valuable strain involves an initial assessment of its phenotypic traits before full-genome sequencing and annotation. Given the absence of comprehensive genomic information, experimentation often involves trial and error, leading to resource-intensive reagent usage. In this context, our strategy involves the initial characterization of the novel *R. erythropolis* strain MGMM8 to identify its genomic genes and clusters. In essence, strain MGMM8 hosts a repertoire of genes and gene clusters that are significant across agricultural, pharmaceutical, and ecological domains. Its resilience, evident through its resistance to xenobiotics and heavy metals, underscores its adaptability, particularly for bioremediation applications in polluted soils and wastewater. Notably, genes associated with quorum-quenching and antibiosis imply potential use in phytopathogen biocontrol, necessitating the establishment of in vitro and in planta experiments for validation. Moving forward, our forthcoming experiments aim to evaluate MGMM8’s biocontrol capabilities and its synergistic potential in conjunction with phytoremediation, assessing its efficacy in remediating both inorganic and organic compounds.

## Figures and Tables

**Figure 1 microorganisms-12-00088-f001:**
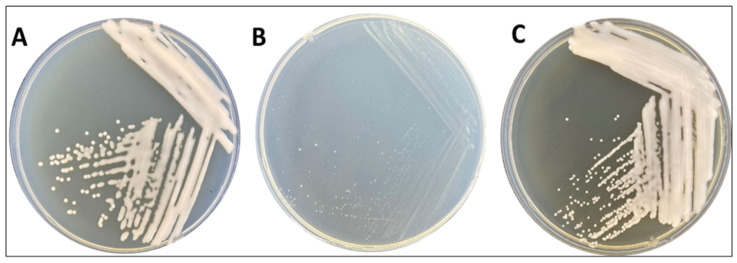
Morphological colonies of *R. erythropolis* MGMM8 on LB media (**A**), Gauze’s synthetic medium no. 1 (**B**), and Gauze’s synthetic medium no. 2 (**C**) after 72 h growth at 30 °C.

**Figure 2 microorganisms-12-00088-f002:**
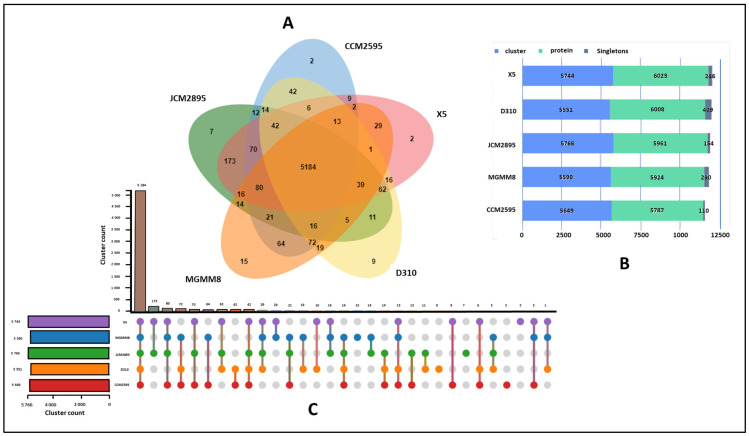
The orthologous protein clusters analysis of *R. erythropolis* strains. Venn diagram showing orthologous cluster distribution among MGMM8 and related species (**A**); counted clusters in each analyzed genome (**B**); and orthologous cluster analysis (**C**).

**Figure 3 microorganisms-12-00088-f003:**
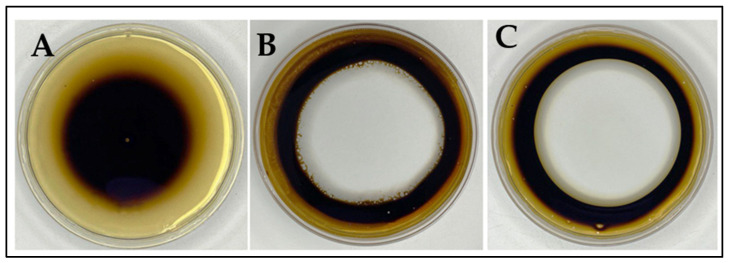
Oil spreading test of *R. erythropolis* MGMM8. Control (**A**); 0.5% SDS solution (**B**); *R. erythropolis* MGMM8 cell-free supernatant (**C**).

**Figure 4 microorganisms-12-00088-f004:**
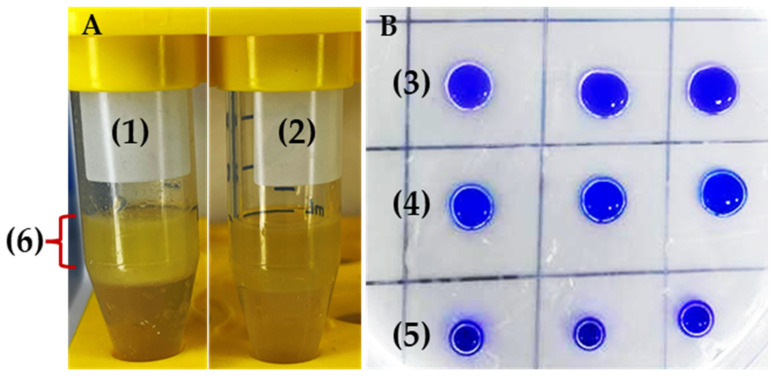
Emulsification (**A**) and drop collapse assay (**B**) of *R. erythropolis* MGMM8. Bio-emulsifier produced by cell-free surfactant of MGMM8 (**1**); control as non-emulsifier (**2**). Cell-free surfactant of MGMM8 (**3**), 1% SDS solution (**4**), and distilled water (**5**) dropped on parafilm—the height of emulsion (**6**) produced after homogenization.

**Table 1 microorganisms-12-00088-t001:** Comparative analysis of secondary metabolite clusters of *R. erythropolis* MGMM8 with closest-related chromosomal genomes from NCBI database.

Type	Most Similar Known Cluster	*R. erythropolis*. Strains. Presence (+) or Absence (−) of Gene Cluster				
		MGMM8	JCM 2895	X5	D310-1	CCM2595
RiPP-like	branched-chain fatty acids	+ (75%)	+ (75%)	+ (75%)	+ (75%)	+ (75%)
LAP	diisonitrile antibiotic SF2768	+ (11%)	+ (11%)	+ (11%)	+ (11%)	+ (11%)
T1PKS	Unidentified	+	+	+	+	+
T1PKS	fulvuthiacene A/fulvuthiacene B	+ (8%)	+ (8%)	+ (8%)	+ (8%)	+ (8%)
NAPAA	ε-Poly-l-lysine	+ (100%)	+ (100%)	+ (100%)	+ (100%)	+ (100%)
NRP-metallophore, NRPS	erythrochelin	+ (57%)	+ (57%)	+ (57%)	+ (57%)	+ (57%)
redox-cofactor	tetronasin	+ (3%)	+ (3%)	+ (3%)	+ (3%)	+ (3%)
NRPS	corynecin III/corynecin I/corynecin II	+ (100%)	+ (100%)	+ (100%)	+ (100%)	+ (100%)
NRPS-like	thiolutin	+ (8%)	+ (8%)	−	−	−
NRP-metallophore, NRPS	heterobactin A/heterobactin S2	+ (100%)	+ (100%)	+ (100%)	+ (100%)	+ (100%)
NRPS, terpene	SF2575	+ (6%)	+ (6%)	+ (6%)	+ (6%)	+ (6%)
NRPS, RRE-containing	coelichelin	+ (27%)	+ (27%)	+ (27%)	+ (27%)	+ (27%)
NRPS	rifamorpholine A/rifamorpholine B/rifamorpholine C/rifamorpholine D/rifamorpholine E	−	+ (3%)	+ (3%)	+ (4%)	−
NRPS	monensin	+ (5%)	+ (5%)	+ (5%)	−	+ (5%)
terpene 3	carotenoid	+ (27%)	+ (27%)	+ (27%)	+ (27%)	+ (37%)
ectoine	ectoine	+ (75%)	+ (75%)	+ (75%)	+ (75%)	+ (75%)
butyrolactone	Unidentified	+	+	+	+	+
PKS-like, amglyccycl	acarbose	+ (7%)	−	−	+ (7%)	+ (7%)
lanthipeptide-class-iii	Unidentified	+	+	+	+	+
NRPS	Unidentified	+	−	−	−	−
NRPS	polyoxin A/polyoxin H	−	+ (5%)	−	−	−
NRPS-like	unidentified	−	−	+	+	+

N.B. (%) – as a similarity percentage to the most known cluster from the Minimum Information about a Biosynthetic Gene cluster (MIBiG) database.

**Table 2 microorganisms-12-00088-t002:** Antimicrobial-resistant genes in the chromosomal genome of *R. erythropolis* MGMM8 compared with the genomes of four selected strains on NCBI database.

ARO Term	AMR Gene Family	Drug Class	Resistance Mechanism	Presence (+) or Absence (−) of Antibiotic-Resistant Genes				
				MGMM8	JCM 2895	X5	D310-1	CCM2595
*VanW* gene in *VanI* cluster	VanW, glycopeptide resistance gene cluster	glycopeptide antibiotic	antibiotic target alteration	+	+	+	+	+
*VanY* gene in *VanB* cluster	VanY, glycopeptide resistance gene cluster	glycopeptide antibiotic	antibiotic target alteration	+	−	−	−	+
*iri*	rifampin monooxygenase	rifamycin antibiotic	antibiotic inactivation	+	−	−	−	+
*RbpA*	RbpA bacterial RNA polymerase-binding protein	rifamycin antibiotic	antibiotic target protection	+	+	+	+	+
Mycobacterium tuberculosis *folC* with mutation conferring resistance to para-aminosalicylic acid	aminosalicylate resistant dihydrofolate synthase	salicylic acid antibiotic	antibiotic target alteration	+	+	+	+	+
Streptomyces venezuelae *rox*	rifampin monooxygenase	rifamycin antibiotic	antibiotic inactivation	−	+	+	+	−

## Data Availability

Data are contained within the article and Appendix A.

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
