# Peer review of "Biotechnological Key Genes of the Rhodococcus erythropolis MGMM8 Genome: Genes for Bioremediation, Antibiotics, Plant Protection, and Growth Stimulation"

_microorganisms, 2023, doi:10.3390/microorganisms12010088_

Round 1
Reviewer 1 Report
Comments and Suggestions for Authors
In the manuscript, the genome was sequenced of a novel strain of Rhodococcus erythropolis, isolated from the rhizosphere of a winter wheat plant. The analysed species has been extensively studied in literature for biotechnological application in several sectors. In this context, the presented analysis must be considered as very preliminary since the presented data lack sufficient innovation, not providing substantial progress on the covered topic. The main important limits of the studied are:
1) in literature and sequence databases, several genomes have been sequenced and characterized. Nevertheless, the novel strain has been compared by the authors with few of those genomes (strains) and the authors did not provide a convincing explanation on this substantial limitation.
2) even if the analysis was intended to provide information on the biotechnological potential of the novel strain, the authors focused on the core genome of R. erythropolis. The employed approach does not allow the authors to define the novel properties and potential applications of the strain as compared to those present in literature.
3) a very limit spectrum of metabolic pathways were analysed without a clear explanation of the criteria adopted by the authors.
4) data are not presented in a readable form without a clear presentation of obtained data in figures.
5) terminology needs revision all over the text.
Comments on the Quality of English LanguageMinor editing of English language required
Author Response
Dear Reviewer,
We are grateful for your consideration of this manuscript and thanks for your keen observation of our manuscript. We appreciate your suggestions, which have been very helpful in improving the manuscript. All the comments that we received on this study have been taken into account and we present our reply to each of them separately.
Kindly find below our response to your comments. All changes for manuscript and figures are in blue font color and strikethrough in red font color for deleted words or sentences.
Two versions of the manuscript are enclosed, one where all the changes have been underlined in red, and a clean version without changes. We have made a considerable effort to take into account the interesting suggestions proposed by the reviewers. In any case, we are open to further comments on our answers.

Reviewer 2 Report
Comments and Suggestions for Authors
The manuscript entitled "Biotechnological key genes of Rhodococcus erythropolis MGMM8 genome: Genes for bioremediation, antibiotics, plant protection, and growth stimulation" is very interesting and provides new discoveries. The "Introduction" chapter effectively introduces the reader to the research topic, which aligns well with the scope of the journal "Microorganisms." The methods used in the research are clearly described in the "Materials and methods" chapter. The results are well presented in the "Results" and Supplementary Materials chapter. The "Disscussion" chapter is also well-constructed.
I suggest accept manuscripts in present form.
Author Response
Dear Reviewer,
We are grateful for your consideration of this manuscript and thanks for your keen observation of our manuscript. We appreciate your comments.
Reviewer 3 Report
Comments and Suggestions for Authors
Dear Authors
Genome mining of potential biotechnology relevant organisms is of high interest. However, just genome analysis is typically not enough to state relevance. You may chose one product and demonstrate production or application of your strain/isolate. Hence I suggest to add an experiment to demonstrate significance!
R. erythropolis strains are well known biosurfactant producers which are supposed to support various important features such as habitat competition, aliphatic degradation, etc.
For genome redundancy there are many more articles online, especially for the degradative power which you may check. There you could add also interesting biocatalysts to your genome mining approach.
minor comments:
- latin phrases should read italics
- keywords should not include same/similar words as in title
-"spp." and "sp." should not read italics. Check throughout the manuscript.
- L95, 96, 161, 192; correct for spaces
- provide reference for Gause media
- L155, 206; italics style "R. erythropolis"
- As and Sb are rather metalloids ... so be more precise using the term "metal"
- The supplemental data could be merged into a single file; add a title page including authors, affiliation, title etc. ... also the quality is not enough; one cannot read any details; these are just copies from antiSMASH.
Comments on the Quality of English Language
not a major issue.
Author Response
Dear Reviewer,
We are grateful for your consideration of this manuscript and thanks for your keen observation of our manuscript. We appreciate your suggestions, which have been very helpful in improving the manuscript. All the comments that we received on this study have been taken into account and we present our reply to each of them separately.
Kindly find below our response to your comments. All changes for the manuscript and figures are in blue font color and strikethrough in red font color for deleted words or sentences.
Two versions of the manuscript are enclosed, one where all the changes have been underlined in red, and a clean version without changes. We have made a considerable effort to take into account the interesting suggestions proposed by the reviewers. In any case, we are open to further comments on our answers.

Round 2
Reviewer 3 Report
Comments and Suggestions for Authors
Dear Authors
well done; now it reads straight and the additional information support the story. There are some other reports on R. erythropolis strains biosurfactants which could be interesting to compare.